# Proving Theorems Recursively

**Haiming Wang**[1][*]   **Huajian Xin**[1][*]   **Zhengying Liu**[2][†]   **Wenda Li**[3]
**Yinya Huang**[4]   **Jianqiao Lu**[5]   **Zhicheng Yang**[6]   **Jing Tang**[6,7]   **Jian Yin**[1][†]
**Zhenguo Li**[2]   **Xiaodan Liang**[1,8][†]
[1]Sun Yat-sen University   [2]Huawei Noahs Ark Lab   [3]University of Edinburgh
[4]CityU   [5]HKU   [6]HKUST (Guangzhou)   [7]HKUST   [8]Pengcheng Laboratory
{wanghm39, xinhj, issjyin}@mail2.sysu.edu.cn, wli8@ed.ac.uk, jqlu@cs.hku.hk
{liuzhengying2, Li.Zhenguo}@huawei.com, yinya.huang@hotmail.com
jingtang@ust.hk {yangzhch6, xdliang328}@gmail.com

## Abstract

Recent advances in automated theorem proving leverages language models to explore expanded search spaces by step-by-step proof generation. However, such approaches are usually based on short-sighted heuristics (e.g., log probability or value function scores) that potentially lead to suboptimal or even distracting subgoals, preventing us from finding longer proofs. To address this challenge, we propose POETRY (PrOvE Theorems RecursivelY), which proves theorems in a recursive, level-by-level manner in the Isabelle theorem prover. Unlike previous step-by-step methods, POETRY searches for a verifiable sketch of the proof at each level and focuses on solving the current level's theorem or conjecture. Detailed proofs of intermediate conjectures within the sketch are temporarily replaced by a placeholder tactic called *sorry*, deferring their proofs to subsequent levels. This approach allows the theorem to be tackled incrementally by outlining the overall theorem at the first level and then solving the intermediate conjectures at deeper levels. Experiments are conducted on the miniF2F and PISA datasets and significant performance gains are observed in our POETRY approach over state-of-the-art methods. POETRY on miniF2F achieves an average proving success rate improvement of 5.1%. Moreover, we observe a substantial increase in the maximum proof length found by POETRY, from 10 to 26.[2]

## 1   Introduction

Neural theorem proving has made significant strides in recent years [Polu and Sutskever, 2020, Han et al., 2022, Polu et al., 2022, Wang et al., 2023c, Jiang et al., 2022a, 2021, 2022b, Wang et al., 2023b, Huang et al., 2024, Thakur et al., 2024, Liu et al., 2023, Xiong et al., 2023], particularly with the integration of language models and search algorithms [Polu and Sutskever, 2020, Han et al., 2022, Jiang et al., 2022a, Yang et al., 2023, Lample et al., 2022]. The combination of language models, which excel at understanding and generating human-like text, and search algorithms, which systematically explore potential solutions, has proven to be a powerful approach to discovering proofs for intricate theorems [Polu and Sutskever, 2020, Lample et al., 2022, Jiang et al., 2022a].

As shown in Figure 1(a), search-based neural theorem proving methods begin with a theorem statement to prove. A formal mathematic environment like Isabelle will first process the theorem statement and provide the initial `proof state`. Starting with the initial proof state, the proving process alternates between sampling new `proof steps` from the language model and obtaining new states

---

[*] These authors contributed equally. [†] Corresponding authors
[2] https://github.com/wiio12/POETRY

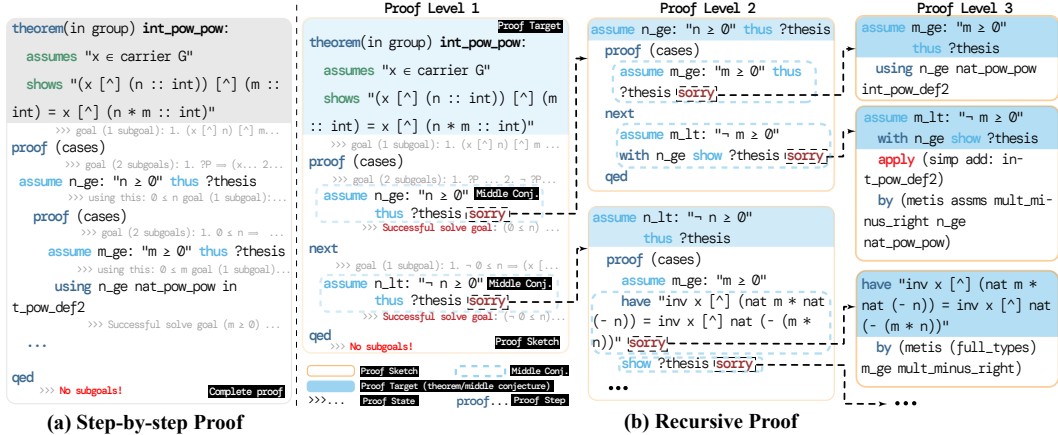

Figure 1: **Comparison between the step-by-step proof and the recursive proof.** (a) A step-by-step proving approach ignores the hierarchical structure inherent in the proof, treating it merely as a sequence of proof steps. The proof cannot be verified as valid until it is fully complete. (b) The recursive proving method decomposes the structured proof into different levels of verifiable proof sketches. Each proof sketch attempts to prove the target theorem or conjecture by outlining the primary steps at the current level and postponing the proof of intermediate conjectures to the next level.

by executing the generated proof steps within the formal mathematic enviroment. Additionally, a search algorithm, such as best-first search or Monte Carlo Tree Search (MCTS), is employed to find a complete path of proof steps. The search algorithm selects the next state to explore based on heuristics such as the log-probability of the proof step [Polu and Sutskever, 2020, Jiang et al., 2022a, Yang et al., 2023], value function scores of the proof state [Han et al., 2022, Polu et al., 2022] (in best-first search), or a PUCT score that combines both [Wang et al., 2023c, Lample et al., 2022] (in MCTS). These heuristics assess the plausibility or potential value of a given step, helping to prioritize the most promising actions. However, these scores are approximate, do not ensure the correctness of the proof direction, and can lead to exploring sub-optimal or distracting subgoals. Even if the language model is capable enough to produce correct proof steps, the search algorithm, guided by short-sighted heuristics, often gets trapped exploring a detailed proof of a meaningless intermediate conjecture. This wastes time and may even cause the algorithm to fail in finding the correct proof path due to a search timeout. Moreover, as the length of the proof increases in more challenging problems, the search space expands exponentially. Consequently, the need for an accurate heuristic to guide the search becomes critical, as a myopic step-by-step approach can easily get lost in the vast expanse of the intermediate proving steps.

To address the aforementioned drawbacks, we propose POETRY, a novel approach that proves the theorem recursively, level by level. As shown in Figure 1(b), POETRY first searches for a *proof sketch*, which is defined to be a verifiable proof outline with the detailed proof of the middle conjecture replaced by a placeholder tactic, *sorry*. The *sorry* tactic signals the formal environment to temporarily ignore the proof of the middle conjecture, assuming it as resolved. Once a validated proof sketch is established, POETRY then endeavors to prove the intermediate conjectures that remain unresolved, also in a recursive, level-by-level manner. This procedure persists until every *sorry* keyword is substituted with a valid proof. Notably, the verified sketch at each level may still contain errors. Since POETRY uses the *sorry* tactic to skip the proof of intermediate conjectures, these conjectures might represent false statements and be unprovable. However, they still serve as correct conjectures to prove the target theorem or conjecture at the current level, resulting in an incorrect proof sketch. For example, to prove the theorem of the commutative property of addition, $a + b = b + a$, a false conjecture such as $a = b$ might be used. However, when actually attempting to prove $a = b$, we would never be able to find a valid proof at the next level. If a false proof sketch is generated and POETRY fails to find the proof for the middle conjecture, it will continue its search to identify a new proof sketch. Nevertheless, Empirical evidence indicates that verifying and ensuring the correctness of the proof sketch at each level before delving into deeper proofs significantly enhances performance. Additionally, we observe a substantial increase in the length of the proofs being able to be generated by POETRY compared with step-by-step approaches. This recursive methodology is inspired by human problem-solving techniques, where complex problems are decomposed into manageable sub-problems, each addressed using a similar recursive strategy.

By adopting this approach, POETRY not only improves the efficiency of its search process but also increases the overall success rate in discovering valid proofs.

We conduct extensive experiments on the theorem proving datasets miniF2F [Zheng et al., 2021] and PISA [Jiang et al., 2021] to validate the effectiveness of our proposed approach. POETRY significantly outperforms previous approaches, achieving a pass rate of $42.2\%$ on both the miniF2F valid and test datasets, respectively. With a $5.1\%$ absolute improvement on average over the previous state-of-the-art. Additionally, our ablation study shows that with recursive theorem proving, we obtain a $3.9\%$ absolute improvement on average compared with step-by-step baselines. Our case study also reveals that POETRY can find proofs substantially longer compared with sequential step-by-step proving methods, the maximum proof length increases from 10 to 26 compared to the step-by-step baseline in the PISA dataset.

## 2 Preliminary

### 2.1 Formal Mathematic Enviroments

We choose Isabelle [Paulson, 1994] as our formal environment. It is widely used for formal verification purposes in academia and industry [Gesellensetter et al., 2008, Klein et al., 2009, Zhang et al., 2024]. It employs a structured proof language called Isar [Wenzel et al., 2004], which facilitates the creation of human-readable proofs and bridges the gap between formal verification and human understanding. As illustrated in Figure 1(a), Isabelle processes each proof step (or tactic) and provides feedback. If the proof step fails to apply, an error message is returned. Otherwise, Isabelle returns a `proof state` along with a special variable, `proof level`, indicating the current level after applying the step. In the Isabelle theorem prover, the `proof level` indicates the depth within a structured proof. This level increases with commands like *have*, *obtain*, and *show*, which introduce new subgoals or conjectures in the proof. Conversely, it decreases with commands like *by*, *qed* and *done*, which conclude a proof block or subgoal.

Isabelle is well-suited for POETRY to accomplish recursive theorem proving. The Isar language is elegantly structured in a level-by-level format, and it contains `proof level` that can be easily used to identify each level. However, the recursive proving technique proposed by POETRY is not specific to Isabelle; the same framework can be extended to other formal mathematical environments like Lean [de Moura et al., 2015], Coq [Barras et al., 1997], and HOL [Harrison, 2009], with additional engineering effort to accommodate the proving strategies. These environments also provide mechanisms to temporarily skip parts of proofs, similar to Isabelle's *sorry* tactic, such as Lean's *sorry*, and Coq's *Admitted*. We will leave the extension of POETRY to other formal environments for future work.

### 2.2 Search-Based Neural Theorem Proving

Search-based neural theorem proving mostly employs the approach introduced by GPT-f [Polu and Sutskever, 2020]. In this method, a pre-trained causal language model predicts the next proof step based on the current proof state and optional context. The language model is trained using data formatted as follows:

$$\begin{aligned} &\texttt{INPUT: CONTEXT \$(context) GOAL \$(proof state) STEP} \\ &\texttt{OUTPUT: \$(proof step)} \end{aligned} \tag{1}$$

where $\$(\cdot)$ represents the substitution operation, and `context` denotes the preceding proof steps leading to the current proof state. At test time, GPT-f employs a best-first search strategy to identify a sequence of proof steps that solve the problem. Specifically, The proof search algorithm constructs a tree-like search structure, where each node represents a proof state and each edge represents a proof step. Starting from the root node, the proof search continuously selects the unexplored node with the highest score and performs an `expansion`. The score for each node is the cumulative log probability of the proof steps that led to the node. During `expansion`, the language model receives the node's proof state and preceding context, then samples $e$ new proof steps. Isabelle subsequently processes these proof steps, generating new proof states or error messages. The search continues until a proof is found or the computational budget is exhausted.

# 3 Methodology

---

**Algorithm 1** Core data curation process

---

1: **function** EXTRACTPROOFSKETCH($proofLines$, $index$)
2:  ▷ $proofLines$: a list of pairs of the format "(proofStep, proofLevel)"
3:  ▷ $index$: the starting index in $proofLines$ for processing
4:  $currentSketch, allSketches \leftarrow$ empty list, empty list
5:  $\_, currentLevel \leftarrow proofLines[index]$            ▷ Obtain the current proof level being extracted
6:  $proofLevel \leftarrow currentLevel$
7:  **while** $proofLevel \geq currentLevel$ **do**  ▷ Extraction ends after the proof level drops below the current proof level
8:    $proofStep, proofLevel \leftarrow proofLines[index]$
9:    $\_, nextLevel \leftarrow proofLines[index + 1]$
10:    **if** $nextLevel = currentLevel$ **then**
11:      $currentSketch$.append($proofStep$)
12:      $index \leftarrow index + 1$
13:    **else if** $nextLevel > currentLevel$ **then**
14:      $currentSketch$.append($proofStep +$ " sorry")         ▷ Replace the next level proof with *sorry*
15:      $deeperSketches, index \leftarrow$ EXTRACTPROOFSKETCH($proofLines, index + 1$)
16:      $allSketches$.extend($deeperSketches$)
17:    **end if**
18:  **end while**
19:  $allSketches$.append($currentSketch$)
20:  **return** $allSketches, index$
21: **end function**

---

## 3.1 Recursive Data Construction

**Proof sketch extraction.** As illustrated in Figure 1(b), to prepare recursive proving data, we need to split theorems into blocks of proof sketches. Each proof sketch focuses solely on the target theorem, conjectures, or subgoals, with the detailed proof of intermediate conjectures or subgoals replaced by the *sorry* tactic. Algorithm 1 presents the pseudocode for the sketch data extraction process. POETRY initially inputs the complete theorem text into Isabelle, which parses it into a sequence of proof lines, containing proof steps and corresponding proof levels. Subsequently, the list of proof lines is passed to the ExtractProofSketch function with the index set to 0, initiating the extraction of all proof sketches. The sketch proof extraction process starts by identifying the current proof level, which is determined by the level of the proof step at the initial index (Line 5). Proof steps that are on the same level as the target theorem, conjectures, or subgoals are those that directly focus on proving the target. Our goal is to retain proof steps with a proof level equal to the current proof level (Lines 10-12) and replace higher-level proof steps with the *sorry* tactic (Lines 13-16). We defer the extraction of higher-level proofs to the recursive call of ExtractProofSketch in Line 15. Finally, the function will return a list of extracted proof sketches, each containing only the current level of proof, as illustrated in Figure 1(b).

**Training data construction.** Following the extraction of proof sketches, POETRY follows Jiang et al. [2022a] and uses PISA, an interactive environment built on top of Isabelle, to extract proof states for each proof step. Subsequently, the proof states and proof steps are reformatted into lines in Equation 1 and used as training examples to fine-tune the language model. Notably, although *sorry* is an independent tactic in Isabelle, POETRY integrates the *sorry* tactic into the preceding proof step (Line 14 in Algorithm 1). This enables the language model to predict the intermediate conjectures and the *sorry* tactic simultaneously. For example, a proof step with the *sorry* keyword would appear as *have "x + 2 = 2x" sorry*. Merging the *sorry* tactic is crucial to ensure that the language model generates proof steps at the current level and postpones higher-level proofs using the *sorry* tactic. Without this merge, the model must determine the use of *sorry* solely based on the context and proof state, which offers no guarantee that the model will generate the necessary *sorry* after stating a conjecture or subgoal. This approach ensures that deep-level proofs are deferred correctly.

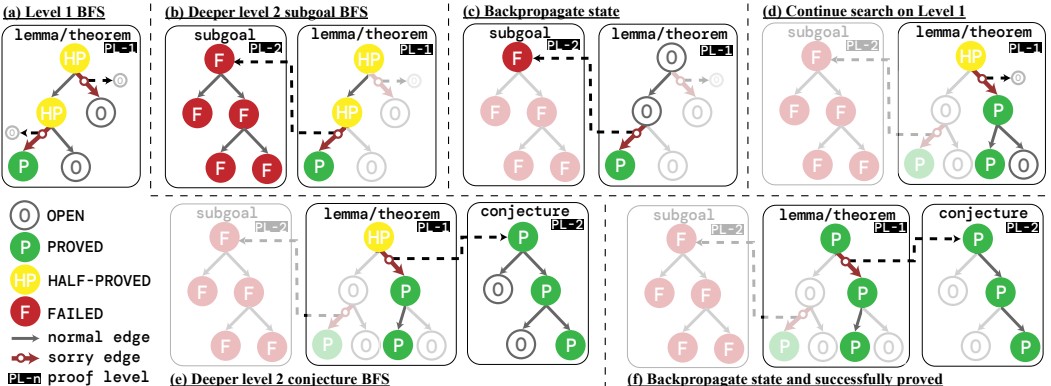

Figure 2: **A walkthrough example of recursive BFS.** Each node in the proof tree is a `proof state` and each edge is a `proof step`. **(a)** The proof search begins by finding the proof sketch at the first level using BFS. The search is paused upon identifying a successful proof path, marked with P and HP nodes. This proof path contains a sorry edge, indicating that it includes skipped conjectures or subgoals that must be addressed in the next level. **(b)** Recursive BFS enters the next level of proof search to attempt to prove the skipped subgoal from the first level. Unfortunately, the proof search for this subgoal fails due to a lack of valid nodes to explore, and the search returns to the first level. **(c)** After the failed attempt to prove the subgoal, the previously established proof path at the first level becomes invalid. Consequently, we backpropagate the failure from the second level's root node up to the first-level root node, updating all the HP nodes to an O node. **(d)** At the first level, with the status set to open for searching proofs, we continued to explore new proof paths. Fortunately, we discovered another proof path. However, this path also contained a sorry edge with a skipped conjecture that needs to be proved at the next level. **(e)** Similar to (b), the recursive BFS proceeds to the next level to search for a proof for the previously skipped conjecture. It successfully finds a proof path without any "sorry" edges (denoted as P nodes), indicating that the conjecture has been proven successfully without any skipped intermediate conjectures or subgoals in the proof path. **(f)** After finding the sub-level proof, the recursive BFS returns to the first level and backpropagates the PROVED message to the root, completing the proof.

## 3.2 Recursive Best-First Search

To prove theorems recursively, POETRY introduces a novel recursive best-first search (recursive BFS) algorithm to conduct a level-by-level proof search. Figure 2 illustrates a complete walkthrough. In general, recursive BFS employs the best-first search technique to search for proof sketches at each level. When a proof sketch is found at a certain level, the algorithm pauses the search at this current level and then proceeds to the next level to solve the skipped middle conjectures by this current level. Once all sketches are found and middle conjectures or subgoals are resolved, a complete proof is achieved. Recursive BFS enhances the generic best-first search to handle multi-level proofs and ensures that the search can pause and continue across different proof levels, adapting BFS to dynamically shift between current and subsequent proof layers based on the progress and outcomes of proof sketches. Below, we will introduce the core elements in the recursive BFS: the `sorry edge` and the node status. For the complete updating rules of nodes status and proof search terminate conditions, please refer to Appendix A.1.

**Sorry edge and node status.** As shown in Figure 2(a), each node in the proof tree is a `proof state`, and each edge is a `proof step`. In a proof state, once a tactic contains a "sorry" keyword (usually after a conjecture or subgoal), we use a special `sorry edge` to connect the parent node and the child node. Then the sorry edge attaches the root node of the next proof level with an unproved conjecture or subgoal. Such root nodes have a score of 0 and will not be selected in the current-level proof search. Moreover, we attach each node in the search tree with one of the status labels: OPEN (the node is open, and no proof has been found so far), FAILED (the node is failed when all potential subproofs or child nodes stemming from it are unable to establish a valid proo), PROVED (the node is proven and part of the successful proof), and HALF-PROVED. A HALF-PROVED node means it belongs to the trajectory that has successfully found a complete proof sketch but contains special `sorry edges` with unsolved next-level subgoal or mid-conjecture. Only after all the mid-conjectures or subgoals in the `sorry edges` from the HALF-PROVED node to the PROVED node are proved will the node be switched to a PROVED node, as illustrated in Figure 2(f).

Using recursive best-first search, POETRY can generate a verifiable proof sketch at each proving level before proceeding to prove the middle conjectures or subgoals at the next level. In essence, POETRY breaks down a lengthy proof into manageable, shorter proof sketches, preventing the

search space from expanding exponentially as the proof length increases. This approach allows search-based methods to find more challenging and longer proofs without necessitating a highly performant value function model to guide the proof search procedure.

# 4 Experiments

## 4.1 Experimental Setup

This section presents our experimental setup, detailing the baselines and evaluation metrics. The implementation details are covered in Appendix A.2.

**Baseline methods.** To fairly compare POETRY with classic step-by-step baselines like GPT-f [Polu and Sutskever, 2020, Jiang et al., 2022a], we implement an Isabelle version of GPT-f, denoted as *GPT-f Baseline*. This baseline model is trained on the same dataset as POETRY, with the only modification being the removal of all *sorry* keywords in the proof steps. All hyperparameters and setups for training and the BFS search are identical to POETRY to ensure a fair comparison.

Notably, the GPT-f Baseline is similar to Thor [Jiang et al., 2022a], except for three main differences. Firstly, GPT-f Baseline does not use Sledgehammer [Paulson, 2010], nor replace the *smt*, *metis* tactic with <hammer> in the proof step for training. Secondly, GPT-f Baseline fine-tunes a 1.3B parameter proofGPT [Azerbayev et al., 2023], whereas Thor uses a 700M model pre-trained on The Pile [Gao et al., 2020]. GPT-f Baseline also uses a newer version of Isabelle which contains more state action pairs for training (detailed in Section A.3). Thirdly, during the proof search, the GPT-f Baseline utilizes the beam-search decoding method instead of sampling to generate proof steps for each proof state.

Aside from the GPT-f Baseline, we also include state-of-the-art search-based neural theorem-proving methods. PACT [Han et al., 2022], FMSCL [Polu et al., 2022], Leandojo [Yang et al., 2023], and COPRA [Thakur et al., 2024] are works focusing on the Lean formal environment.[3] Contrastively, Thor [Jiang et al., 2022a], Thor with expert iteration on auto-formalized data [Wu et al., 2022] and Thor + Magnushammer [Mikuła et al., 2023] are works done in Isabelle. Moreover, for methods with LLMs, COPRA is an in-context learning agent that uses GPT-4 [OpenAI, 2023] to generate proof steps and prove the theorem step by step.

We do NOT compare our methods with LLM-based proving approaches like DSP [Jiang et al., 2022b], Lyra [Zheng et al., 2023], or LEGO-Prover [Wang et al., 2023b]. These approaches employ general-purpose large language models (LLMs), such as ChatGPT or GPT-4, which feature several orders of magnitude more parameters than the models considered in our study. Moreover, these methods typically utilize proofs in natural language to guide the generation of formal code without searching and attempting to solve each problem 100 times. In contrast, POETRY provides a performance evaluation at pass@1, attempting to prove the theorem once for each problem.

**Evaluation datasets and metrics.** For evaluation, we use two datasets, the miniF2F dataset [Zheng et al., 2021], and the PISA [Jiang et al., 2021]. The miniF2F dataset comprises 488 problems with varying levels of difficulty, ranging from basic algebra and number theory, originating from the MATH dataset [Hendrycks et al., 2021], to more challenging problems found in the AIME[4] and IMO [Daniel Selsam, 2019]. The problems are divided into valid and test sets, with 244 problems each. The miniF2F dataset only contains problem statements and we only evaluate our method on this dataset, without any training. The other dataset we adopt is the PISA test set, which comprises theorems from the Archive of Formal Proofs [MacKenzie et al., 2021] and the Isabelle standard library [Nipkow et al., 2002]. To better understand how POETRY performs in complex problems with multiple levels, we subdivided the test set into two subsets: single-level and multi-level. The PISA single-level subset contains problems with only one level in the ground truth human-written proofs, whereas the PISA multi-level subset includes problems with multiple proof levels. A more comprehensive analysis of the PISA dataset is shown in Appendix A.3. For evaluation metrics, we follow Jiang et al. [2022a], Yang et al. [2023] and use pass@1 as the evaluation metric, where each

---

[3]HTPS [Lample et al., 2022] achieve 57% and 41% on miniF2F valid and test set for pass@64, but they didn't provide the pass@1 results. Additionally, the model is fine-tuned on the miniF2F-valid with online training, which is not a fair comparison with POETRY.

[4]https://artofproblemsolving.com/wiki/index.php?title=AIME_Problems_and_Solutions

Table 1: **Comparing with baseline.** The table displays the pass@1 success rates of the baselines and POETRY, The highest success rates for each set are highlighted in bold.

| Success rate | miniF2F-valid | miniF2F-test | PISA | single-level | multi-level |
|---|---|---|---|---|---|
| Thor w/o sledgehammer | 25.0% | 24.2% | 39.0% | - | - |
| GPT-f Baseline | 39.3% | 37.3% | 49.0% | **65.5%** | 11.1% |
| − with sampling decoding | 30.3% | 31.5% | 43.2% | 57.8% | 9.8% |
| POETRY | **42.2%** | **42.2%** | **49.7%** | 65.4% | **13.6%** |

Table 2: **Comparing with state-of-the-art search-based methods on the miniF2F dataset.** The table displays the pass@1 success rates of previous works and POETRY, The highest success rates for each set are highlighted in bold. [6]

| Success rate | environment | miniF2F-valid | miniF2F-test |
|---|---|---|---|
| *Baselines* | | | |
| PACT [Han et al., 2022] | Lean | 23.9% | 24.6% |
| Leandojo [Yang et al., 2023] | Lean | - | 26.5% |
| FMSCL [Polu et al., 2022] | Lean | 33.6% | 29.6% |
| COPRA  [Thakur et al., 2024] | Lean | - | 30.7% |
| Thor [Jiang et al., 2022a] | Isabelle | 28.3% | 29.9% |
| Thor + expert iteration [Wu et al., 2022] | Isabelle | 37.3% | 35.2% |
| Thor + Magnushammer [Mikuła et al., 2023] | Isabelle | 36.9% | 37.3% |
| *Ours* | | | |
| POETRY | Isabelle | **42.2%** | **42.2%** |

theorem in the dataset is proved once by POETRY. Then we calculate the proportion of the theorems being successfully proven.

## 4.2   Main Results

**Comparison with language model-only baselines.** As shown in Table 5, we compare POETRY with baselines that only utilize language models to search for proofs. Thor w/o sledgehammer is the language model-only version of Thor [Jiang et al., 2022a]. It does not call the sledgehammer during the proof search. Our reproduced GPT-f Baselines outperform Thor w/o sledgehammer by 13.7% in miniF2F and 10.6% in the PISA test set. This performance boost is mostly due to using the beam-search decoding strategy during the proof search, as we observe the performance of the GPT-f Baseline with sampling drops by 6.8% compared with the beam-search version. This is because the beam-search decoding method is guaranteed to produce $e$ different proof steps for each proof state, whereas the sampling will produce duplicate proof steps, making the actual number of proof steps generated per expansion smaller than $e$. The remaining performance improvements are mostly contributed by larger model sizes and better pertaining.

Compared with the GPT-f Baseline, we can observe the benefit of the recursive theorem proving. POETRY outperforms GPT-f Baselines by 3.9% in the miniF2F dataset on average, and 0.7% in the PISA test set. The modest performance gain observed in the PISA test set is primarily attributed to the skewed distribution of problem complexity, with the majority of problems containing only a single proof level (see Table 3). POETRY executes nearly identically to the GPT-f Baseline when encountering proofs with only one level, resulting in similar performance within the single-level subset. In contrast, POETRY achieves a 2.5% improvement on the multi-level subset. Furthermore, POETRY solves a very distinct set of theorems compared with GPT-f Baseline in PISA, with 99 out of 1109 theorem solved by POETRY can not be proved by GPT-f Baseline, taking up 4.4% in total. This outcome well supports the effectiveness of our proposed recursive proving method. Moreover, the gap between step-by-step approaches and POETRY does not end here. The effectiveness of POETRY will become even more pronounced as the language models are continuously improved and solve more complex proofs, where the bottleneck caused by searching comes to the fore yet POETRY is demonstrated effective for searching.

---

[6]The performance of methods in Lean and Isabelle cannot be directly compared, as the training data differ fundamentally in coding language or data volume. However, despite these differences, the high-level concepts behind these approaches share significant similarities.

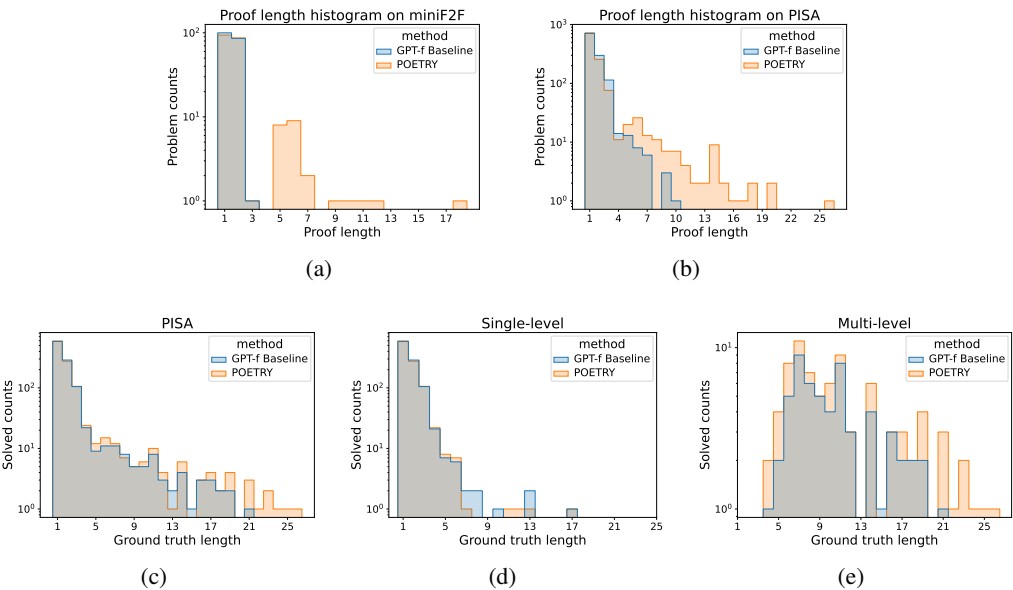

(a)

(b)

(c)        (d)        (e)

Figure 3: **(a)&(b) Proof length comparison between POETRY and GPT-f Baseline.** The y-axis is shown in the log scale. (a) Proof length's histogram of found proof in miniF2F dataset. most of the proof found is within 3 steps long, especially for GPT-f Baselines, but POETRY managed to find longer proof up to 18 proof steps in one proof. (b) Proof length's histogram of found proof in the PISA dataset. POETRY discovers a lot more proofs with longer proof lengths. **(c), (d) and (e) Number of problems solved by GPT-f Baseline and POETRY based on the length of the ground truth proofs.** (a) The distribution in the entire PISA dataset. POETRY has a clear tendency to solve problems with longer ground truth proofs. (b) The distribution in the single-level subset. In this subset, POETRY and GPT-f Baseline should behave nearly identically; therefore, there are not many differences as expected. (c) The distribution in the multi-level subset. POETRY has a distinctive advantage over GPT-f Baseline. This is the subset where all the complex theorems exist.

**Comparison with state-of-the-art methods.** In Table 2, we illustrate the proportion of successful proofs found on the miniF2F dataset. Due to the larger amount of formal data, as well as the help of hands in ATP like the sledgehammer, the approaches using Isabelle tend to achieve a higher pass rate compared with approaches using Lean environments. Our proposed POETRY significantly outperforms all such approaches. POETRY outperforms Thor+Magnushammer by $5.1\%$ on average, the highest performance on the miniF2F dataset with the search-based method at pass@1.

Notably, the recursive proving method is orthogonal to these baseline approaches. It can be further improved with the use of Sledgehammer or Magnushammer [Jiang et al., 2022a, Mikuła et al., 2023], running expert iteration on the training set [Polu et al., 2022, Wu et al., 2022], using retrieval augmented proof step generation techniques [Yang et al., 2023], or even better search algorithm in each level [Wang et al., 2023c, Lample et al., 2022]. As these are not the focus of the current paper, we leave the integration for future work.

### 4.3 Analysis

**Can POETRY find longer proof?** Figure 3 (a) and (b) compares the proof length of proofs discovered by the GPT-f Baseline and POETRY in both the miniF2F dataset and the PISA test set. It can be observed that the proof lengths found by POETRY are longer than those found by the GPT-f Baseline. The average proof length increases from $1.47$ to $2.13$ in the miniF2F dataset and $1.62$ to $2.09$ in the PISA test set. Prominently, the maximum proof length increases from $3$ to $18$ compared with the GPT-f Baselines in the miniF2F dataset, and from $10$ to $26$ in the PISA test set. This proof length is unattainable without a recursive proving method. By comparison, the maximum proof length found by Leandojo in the miniF2F test set is $4$, with an average proof length of $1.35$. Therefore, it's evident that POETRY expands the possibility of discovering longer proofs and addressing more challenging problems.

**Can POETRY find harder proof?** The ability to find longer proofs does not necessarily imply the ability to find harder proofs. We analyze this issue from two perspectives. First, POETRY does not generally produce longer proofs than the GPT-f Baseline for problems both approaches successfully solve. Algorithmically, POETRY operates almost identically to the GPT-f Baseline,

```
lemma(in UP_cring) n_mult_closed:
  assumes "f ∈ carrier P"
  shows "n_mult f ∈ carrier P"

proof(rule UP_car_memI[of "deg R f"])
  show "∧n. deg R f < n ⟹ n_mult f n = 0"
    unfolding n_mult_def
    using assms
    unfolding P_def
    by (simp add: UP_car_memE(2))      Proof level 2
  show "∧n. n_mult f n ∈ carrier R"
    using assms
    unfolding n_mult_def
    by (simp add: assms cfs_closed) Proof level 2
qed
                                       Proof level 1
```
(a)

```
lemma(in UP_cring) n_mult_closed:
  assumes "f ∈ carrier P"
  shows "n_mult f ∈ carrier P"
proof(rule UP_car_memI[of "deg R f"])
  fix n
  assume A: "deg R f < n"
  show "n_mult f n = 0"
    unfolding n_mult_def
    proof -
    ✗ Timeout after 600 seconds
```
Path 1

```
proof(rule UP_car_memI[of "deg R f"])
  show "∧n. deg R f < n ⟹ n_mult f n = 0"
    ✗ Never explored
```
Path 2

(b)

Figure 4: **Case comparison between POETRY and GPT-f Baseline.** (a) Recursive proof found by POETRY in **71.2** seconds, the proof contains two proof levels. (b) Failure-proof paths found by the GPT-f Baseline. GPT-f Baseline failed to find proof due to timeout after **600** seconds. We select two different failure proof paths found by GPT-f Baseline.

only progressing to deeper levels when it identifies a complete proof sketch. Therefore, we expect similar proof lengths for problems solved by both methods. Statistically, 82.3% of the problems solved by both methods have identical proof lengths, and 96.0% have proof length differences of less than 3 steps, attributable to algorithmic randomness. Occasionally, POETRY generates longer proofs with redundant steps, as shown in Figure 6 in the Appendix. This is due to POETRYs greedy exploration mechanism, which sometimes explores dummy sketches. These cases are rare (2.4% of solved problems where POETRYs proof is 3 steps longer than GPT-f Baselines). We believe this issue can be addressed by implementing a value function to prioritize informative sketches over redundant ones in future work.

Secondly, we included histograms displaying the number of problems categorized by the lengths of the ground truth proofs (Figure 3 (c), (d), and (e)). From these figures, it is evident that POETRY shows a marked tendency to solve harder problems (i.e., those with longer ground truth proofs) and consistently outperforms the GPT-f Baseline across various levels of problem difficulty within the multi-level subset. Therefore, we conclude that POETRY demonstrates a distinct advantage over the GPT-f Baseline, showcasing its ability to solve more complex problems.

**Case study.** As illustrated in Figure 4, we compare the proof found by the POETRY with the failed attempts by the GPT-f Baseline. The theorem n_mult_closed states that if a polynomial $f$ belongs to the carrier set of polynomials $P$, then the operation n_mult applied to $f$ results in a polynomial that also belongs to $P$. As shown in Figure 4(a), the proof found by the POETRY contains two levels, marked with different shades of blue. The first level is completed by first showing two main properties: (i) Zero polynomial condition (the first show statement in Line 2): For any integer $n$ greater than the degree of $f$, n_mult $fn$ must be zero. (ii) Closure under carrier (the second show statement in Line 7): For any integer $n$, the result n_mult $fn$ must be within the carrier set $R$. When proving the first level, the detailed proof of these two properties will be skipped with the sorry tactic. After the first level of the proof has been found, POETRY searches for the proof of these properties one by one in the next level. In contrast, the GPT-f Baseline failed to find valid proof for this problem, resulting in a search timeout after reaching 600 seconds of time limit. Two failure search trajectories are selected and shown in Figure 4(b). For proof path 1, the proof searches astray and tries to utilize a more complex way to prove the first property, resulting in a timeout. The GPT-f Baseline also identified the first two steps in POETRY's proof. However, this proof path never had the chance to be further explored before the timeout occurred. From this case, we can see that by recursively proving the theorem, the proof with 11 steps is broken down into 3 proof sketches with a maximum length of 4. Therefore, POETRY effectively prevents the proof search from wasting too much time on searching for useless mid-step conjectures. These are typical successful cases for POETRY; additional failure cases can be found in Section A.4 of the Appendix.

## 5 Related Works

**Search-based neural theorem proving.** Our work is closely related to prior work on step-by-step search-based nerual theorem proving. GPT-f [Polu and Sutskever, 2020] is the first to apply transformer-based language models to generate single-step action for theorem proving in Metamath. With the ability to generate arbitrary proof text, modern ATPs advance drastically and are capable of proving theorems in complex ITPs like Lean [de Moura et al., 2015] or Isabelle [Paulson,

1994]. The follow-up work PACT [Han et al., 2022] proposes auxiliary pre-training tasks for action-generating language models. Polu et al. [2022] uses expert iteration and syntactic data to bootstrap the language model's performance. Most recently, HTPS [Lample et al., 2022] plugs in Monte-Carlo Tree Search [Silver et al., 2016] in this framework and applies an online version of expert iteration. DT-Solver [Wang et al., 2023c] improves HTPS by enabling backtracking during proof search, increasing the robustness. LeanDojo [Yang et al., 2023] retrieve possible premise to assist the generation of a single proof step. Lisa and Thor [Jiang et al., 2021, 2022a] tackle theorem proving in Isabelle, which combines traditional ATPs and language models to suggest proof steps, in a neuro-symbolic way. All theorem-proving method introduced above proves theorems step-by-step, with short-sighted heuristics guiding the search to find a correct proof path.

**Nerual theorem proving with a large language model.** Another popular paradigm for automated theorem proving resorts to large pre-trained language models for proof context generation in an in-context-learning manner, without finetuning on formal mathematic datasets. DSP [Jiang et al., 2022b] uses OpenAI Codex LLM [Chen et al., 2021] to generate the entire proofs guided by informal proof. It suffers from hallucination problems with LLM and requires multiple attempts for each problem to ensure correctness. Lyra [Zheng et al., 2023] improves on DSP and uses GPT-4's auto-correction ability to correct previous error attempts. Baldur [First et al., 2023] also uses Minerva [Lewkowycz et al., 2022] for whole proof generation using the initial theorem statement. To prevent hallucination, Baldur finetunes a small model that uses error messages to correct the generated faulty proof. MUSTARD [Huang et al., 2024] generates the problem and the solution simultaneously with ChatGPT and uses Lean as a verifier to check the correctness of the generated content.

**Subgoal-based AI agents.** Another domain that is closely related to our paper is subgoal-based AI agents [Wang et al., 2023b,a, Wei et al., 2023]. These agents decompose the major tasks into small sub-objectives and tackle them one by one. However, most AI agents do not focus on formal mathematic problems, which require compiling the rules of formal environments. LEGO-Prover [Wang et al., 2023b] approaches the theorem-proving problem by decomposing the target into sub-goal lemmas and building the proof block by block. However, not all the subgoals can be easily decomposed into lemmas. Many mid-conjectures or subgoals are specific to the current problem and involve shared variables defined in the previous proving process, making them unsuitable for extraction as lemmas, or sometimes impossible to extract as lemmas. kSubS [Czechowski et al., 2024] utilizes a subgoal generation model to produce mid-step proof states and employs a policy model to generate paths in between. However, the generated proof must adhere to the generated proof states, thus the method cannot be applied to more complex real-world datasets like miniF2F. Moreover, the proposed subgoal generator constrains the ability of the policy model to explore freely and find solutions beyond predefined subgoals.

# 6 Limitations

The proposed method proves theorems recursively by producing a verifiable proof sketch at each level. Although this leads to consistent performance improvements, there is no theoretical guarantee that it will avoid the problem of infinite action space for each proof step generation and the problem of exponential search space with respect to the depth of the search tree. Furthermore, applying the framework of POETRY to other formal languages such as Lean or Coq is straightforward but would require a non-neglectable amount of engineering efforts on some language-specific aspects.

# 7 Conclusion

In this work, we introduce a novel theorem-proving method, POETRY, which recursively proves the theorem in a level-by-level manner. POETRY searches for a verifiable proof sketch in each level, focusing on proving the target theorem, conjecture, or subgoals in the current level, and utilizes a special *sorry* tactic to defer detailed proofs of mid-conjectures or subgoals. POETRY introduces a fundamentally different theorem-proving paradigm to the community, preventing short-sighted proof searches that easily go astray. The recursive dataset decomposes long proofs into short proof sketches within a tractable search space. Extensive experiments show that POETRY can indeed improve the pass rates on the miniF2F dataset and PISA test set, and can find longer proofs compared to step-by-step approaches.

## Acknowledgements

This work is supported in part by the Major Key Project of PCL (No. PCL2024A04), National Natural Science Foundation of China (U2001211,U22B2060), Research Foundation of Science and Technology Plan Project of Guangzhou City2023B01J00012024B01W0004, National Science and Technology Major Project (2020AAA0109704), National Science Foundation of China Grant No. 62476293, Guangdong Outstanding Youth Fund (Grant No. 2021B1515020061), Shenzhen Science and Technology Program (Grant No. GJHZ20220913142600001), Nansha Key RD Program under Grant No.2022ZD014.

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

# A  More Details on POETRY

The outline of the Appendix A is as follows:

- More details on our proposed recursive best-first search.
- Implementation details for POETRY, including the hyperparameters for the methods and machine configuration.
- More details on the newly extracted PISA dataset, and additional analysis of the statistics and characteristics of the dataset.
- Additional results include a comparison table with LLM-based approaches, input/output comparisons for GPT-f Baseline and POETRY, and an analysis of POETRYs failure cases.

Appendix B discusses the broader impacts of POETRY.

Appendix C includes more examples of found theorems by POETRY.

## A.1  Details on Recursive BFS

In this section, we discuss more details on the recursive BFS algorithm. Section 3.2 only introduces the overall process of how recursive BFS runs, and no detailed introduction on status update rules, pause and continue of the recursive BFS and terminate conditions. We discuss each in detail below.

**Status update rules.** The status update happens whenever a node finishes its expansion, which adds all the newly created nodes as children. The update will propagate from the expanded node all the way to the root node. A node's status will be marked as FAILED if all the children are FAILED, and a node will be marked as PROVED or HALF-PROVED if any of its children is PROVED or HALF-PROVED. Additionally, when POETRY encounters a `sorry` edge there exists special update rules. If a node is connected to a PROVED node with a sorry edge, and the next level root node is OPEN, this means the mid-conjectures/subgoals represented by the sorry edge have not been proved, the node will be marked as HALF-PROVED (Figure 2(a)). As illustrated in Figure 2(c), if the sub-root node has failed, the original HALFPROVE status of the node will be updated to OPEN. And if the sub-root node is PROVED, the original HALFPROVE status of the node will be marked as PROVED (Figure 2(f)).

**Pause and continue of recursive BFS.** Figure 2(a)-(d) illustrates the pause and continue of recursive BFS. During the proof search, whenever a proof sketch is found (Figure 2(a)), the current level of the best-first search will be paused, and POETRY will find the **last** unproved sorry edge and recursively call the best-first search algorithm to find the proof for the next-level root node attached in the sorry edge (Figure 2(b)). If the next-level best-first search fails to find proof for the sub-root node (The status of the sub-root node is marked as FAILED), POETRY will update the status of the search tree and continue the paused best-first search for the current level and try to find new proof sketches (Figure 2(c)-(d)).

**Terminate conditions.** The proof search of the current level will terminate and return to the upper-level proof search under these scenarios: 1) A complete proof for this level is found, which means all the middle conjectures or subgoals have been proven by the deeper levels of proof searches, recursively. The root node status of the current level proof search is marked as PROVED. 2) For proof search in a level higher than 1, a timeout of 120s has been reached. The root node status will be marked as FAILED. 3) All the nodes in the proof tree have been explored and no proof has been found, the root node status will also be marked as FAILED. Additionally, a global timeout of 600s is added to the entire recursive BFS, ensuring each theorem will not be searched longer than 5 minutes. We can finally obtain complete proof for the target theorem after the first level best-first search return as proved, as shown in Figure 2(f).

## A.2  Implementation Details

In this work, we use a decoder-only transformer [Devlin et al., 2019] architecture pre-trained with proof-pile v1.1 dataset [Azerbayev et al., 2023], with 1.3b parameters, 24 layers, 16 attention heads, a hidden dimension of 2048, and a GPT-2 tokenizer with 50400 vocabulary size. We use the alpaca[7]

---

[7]https://github.com/tatsu-lab/stanford_alpaca

Table 3: **Dataset statistics.** The table displays the dataset statistics for our newly extracted PISA dataset based on Isabelle 2022.

|  | train | valid | test | single-level | multi-level |
|---|---|---|---|---|---|
| Number of theorems | $236,978$ | $2,347$ | $2,236$ | $1,558$ | $681$ |
| Number of proof steps | $3,018,407$ | $27,419$ | $27,653$ | $3,982$ | $23,671$ |
| Average proof length | 12.7 | 11.7 | 11.8 | 2.4 | 33.5 |
| Maximum proof length | $10,320$ | $1,236$ | $1,079$ | 204 | 1079 |
| Average proof level | 1.5 | 1.5 | 1.5 | 1.0 | 2.6 |
| Maximum proof level | 26 | 9 | 10 | 1 | 10 |

codebase for finetuning the model on our recursive dataset. During fine-tuning, we use a global batch size of 256 with 3500 steps of warmup using the AdamW optimizer. We use the cosine scheduling strategy with a maximum learning rate of $3e-4$ and a minimum learning rate of $3e-5$. Our model is finetuned with $100,000$ steps training budgets and inferences using the lowest validation loss checkpoints with early stopping.

For the configuration of recursive best-first search. We use a global timeout of 600 seconds; each proof step has a timeout limit of 10 seconds. The number of samples per expansion $e$ is set to 32, and we use beamsearch decoding strategies to sample proof steps from the language model. The maximum number of steps for expansion is set to 128, and the maximum recursive depth for searching deeper level proof is set to 10. For proof searches other than the first level, a local timeout of 120 seconds is also applied.

**Machine configuration.** We use Nvidia A800 GPU with 80GB of GPU memory for fine-tuning. The training server has 104 CPU cores and 1024GB of CPU memory. The finetuning takes around 100 GPU hours and requires an additional 50 GPU hours to run a single evaluation on the miniF2F test set.

## A.3 Dataset Details

In this section, we further discuss the details of our newly extracted PISA dataset, including the dataset statistics and other interesting aspects of the dataset.

**Dataset Statistics.** We follow Jiang et al. [2021, 2022a] and extract data from Isabelle 2022, as well as the corresponding version of the Archive of Formal Proof library.[8] We provide detailed statistics for our fine-tuning dataset. As shown in Table 3, the newly constructed PISA dataset contains 3.02 million proof steps in the training data. In contrast, the old PISA dataset extracted by LISA [Jiang et al., 2021] only contains 2.49 million proof steps. Another interesting factor of the dataset statistics is the two subsets of the PISA test. The single-level test set contains 2/3 of the problems in the test set, but only $14\%$ of the proof step. In contrast, the multi-level subset contains the remaining $86\%$ proof steps.

**How recursive the dataset is?** As illustrated in Figure 5(a), the figure shows the histogram of the number of proof levels in a single theorem against the number of theorems. As expected most of the theorem in the training dataset only contains one proof level, which does not require recursive proving at all. This result matches the Pareto principle [Dunford et al., 2021] where the majority of the problems are simple and could be tackled without the recursive proving technique. However, it's the challenging problems that are of most interest to us, where they can test the boundary of our method's actual proving ability.

**How much does the search space shrink by proving the theorem recursively?** As the verified proof sketches might not always be correct due to mid-conjectures/subgoals' proof being skipped by sorry, we can not accurately calculate the search space is shrunk to which extent. However, we can have a lower bound search space calculated using ground truth proof. Figure 5(b) shows the histogram of the number of proof steps that need to be completed until a proof/proof sketch can pass the verification of Isabelle against the number of these proofs. For conventional step-by-step approaches, the proof is the original one, and for POETRY, the proof is a proof sketch. We can

---

[8]The original dataset is extracted with Isabelle 2021, resulting in x fewer theorems and x fewer lines of state action pair.

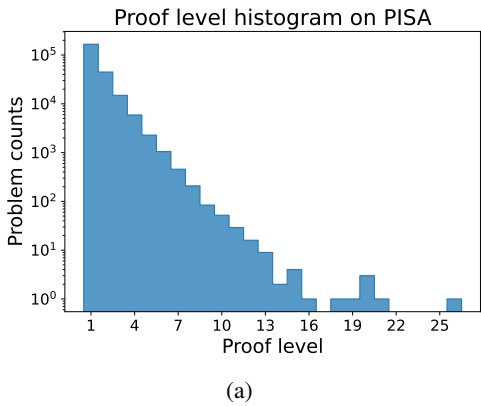
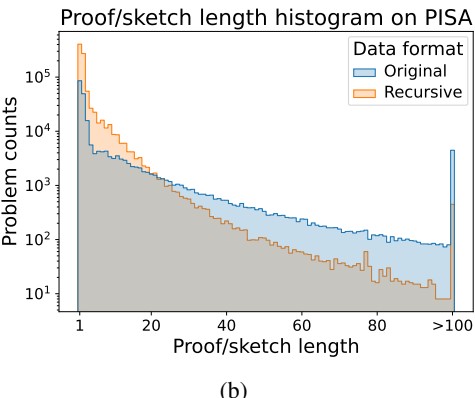

|        (a)        |        (b)        |

Figure 5: **Distribution of proof level and proof length in PISA dataset.** (a) Histogram of proof level in the PISA training set. The maximum proof level can reach 26 (b) Comparison between the number of steps in the original proof and the extracted proof sketches. By breaking the original proof into proof sketches, the proof length is reduced substantially.

Table 4: **Proving Success Rates on the miniF2F Dataset with Isabelle.** The table displays the success rates of previous works using large language models. These approaches are not directly comparable with POETRY but are listed here only for demonstration.

| Success rate | LLM | miniF2F-valid | miniF2F-test |
|---|---|---|---|
| Draft, sketch, and Prove | Codex | 42.6% | 39.3% |
| Subgoal-Learning | ChatGPT | 48.0% | 45.5% |
| LEGO-Prover (model informal proof) | ChatGPT | **52.0%** | **45.5%** |
| POETRY | proofGPT-1.3B | 42.2% | 42.2% |

observe that the proof length of POETRY is substantially shifted towards a shorter proof length per proof. On average, there are 3.3 proof steps for POETRY and 12.7 proof steps for step-by-step baseline per proof. And that would be $32^{9.4}$ times smaller the search space per proof on average with $e = 32$.

## A.4 Additional results

**Comparison with LLM-based approaches.** For a more comprehensive view of the relationship between POETRY and other state-of-the-art methods, we include Table 4 to illustrate the performance in miniF2F using the large language model like ChatGPT [OpenAI, 2023]. It is important to note that these methods use a completely different approach to prove theorems. Instead of performing searches using BFS algorithms, these approaches leverage the general-purpose capabilities of LLMs to translate natural language proofs into formal code (often termed autoformalization). Therefore, comparing these methods with POETRY is not only unfair in terms of model size and the amount of training data but also incomparable in core methodology.

**Input and output comparison for GPT-f Baseline and POETRY.** We also provide a table illustrating the detailed input and output pair constructed for language model training. This is the same theorem example shown in Figure 2. The final training data for POETRY and GPT-f Baseline have the same input text and the same number of training examples. The only difference is the additional sorry keyword in proof steps that state conjectures or subgoals. Therefore, the model trained with POETRY data does not receive more information compared to the GPT-f Baseline. The success of POETRY comes from the data curation process, the novel recursive BFS, and the recursive proving methodology itself. We will include this table in our paper for better illustration.

**Failure cases of POETRY** We provide additional failure cases of POETRY to better understand the limitations of current recursive proving approaches. Compared to the GPT-f Baseline, a notable issue with the recursive framework in POETRY stems from its greedy exploration strategy. POETRY aggressively explores deeper-level proofs as soon as it finds a valid proof sketch, yet sometimes

Table 5: **Input and Output Comparison for GPT-f Baseline and POETRY on the Training Data.** Both GPT-f Baseline and POETRY have the exact same input state and number of training examples. The only differences are in lines 2, 6, and 21, where a `sorry` is appended to the end of the tactic.

| # | Input States | GPT-f Baseline Output | POETRY Output |
|---|---|---|---|
| 1 | `goal (1 subgoal): 1. (x [^] n) [^] m...` | `proof (cases)` | `proof (cases)` |
| 2 | `goal (2 subgoals): 1. ?P -> (x [^] n) ...` | `assume n_ge: "n \<ge> 0" thus ?thesis` | `assume n_ge: "n \<ge> 0" thus ?thesis` `sorry` |
| 3 | `goal (2 subgoals): 1. 0 <= n -> ?P1...` | `using n_ge nat_pow_pow` | `using n_ge nat_pow_pow` |
| 4 | `using this: 0 <= m 0 <= n (x [^] nat n...` | `by (simp add: mult_less_0_iff)` | `by (simp add: mult_less_0_iff)` |
| 5 | `Successful attempt to solve goal by...` | `next` | `next` |
| 6 | `goal (1 subgoal): 1. 0<n -> ň0 <= m...` | `assume m_lt: "..." show ?thesis` | `assume m_lt: "..." show ?thesis` `sorry` |
| 7 | `using this: 0 <= n ň 0 <= m goal (1 ...` | `apply (simp add: int_pow_def2)` | `apply (simp add: int_pow_def2)` |
| 8 | `goal (1 subgoal): 1. 0 <= n -> ň 0 ...` | `by (metis assms mult_minus_right)` | `by (metis assms mult_minus_right)` |
| ... | ... | ... | ... |
| 21 | `goal (1 subgoal): 1. ň 0 <= n -> ň0 <...` | `assume m_lt: "..." thus ?thesis` | `assume m_lt: "..." thus ?thesis` `sorry` |
| 22 | `proof (prove) using this: ň 0 <= m ...` | `using n_lt` | `using n_lt` |
| 23 | `proof (prove) using this: ň 0 <= m ň 0...` | `by (auto simp: int_pow_def2)` | `by (auto simp: int_pow_def2)` |
| 24 | `Successful attempt to solve goal by..` | `qed` | `qed` |
| 25 | `goal: this: (x [^] n) [^]... No subgoals!` | `qed` | `qed` |

(a)  (b)  (c)

Figure 6: **Failure cases of POETRY in Proof Search.** (a) A case where POETRY finds a longer proof compared to GPT-f Baseline. Due to the greedy exploration of successful proof sketches, POETRY might find proofs with redundant wrapping of proof steps, comprising 5 steps, whereas GPT-f Baseline finds proofs with 2 steps and less redundancy. Note that this happens very rarely (2.4% of both solved problems). (b) and (c) A failure case where POETRY fails to find proof due to greedy exploration of successful sketches. The correct proof is shown in (b), where the correct proof steps remain on the same level after the tactic `apply (induct n)`. However, POETRY finds a sketch in advance and greedily explores deeper sketches of the last `sorry`, eventually failing to find the correct proof due to a timeout.

these sketches are dummy sketches, lacking meaningful proving steps (e.g., a general deduction framework as shown in Figure 6 (a)). In other cases, the correct solution exists at a shallow proof level but is bypassed, never returning from deeper exploration (Figure 6 (b) and (c)). Introducing a value function to prioritize informative sketches and selectively deepen or retain the current level of exploration could address these issues.

# B  Broader Impact

The research presented in this paper has the potential to advance automated theorem proving, AI for Math and software engineering. The advancement can enhance the capabilities of large language models in formal theorem proving, contributing to more reliable mathematical proof verification and providing valuable educational resources for students and researchers. By directly releasing the code, model and data, we aim to ensure the responsible use of our work, fostering further innovation and maintaining high standards of data privacy and intellectual property compliance.

We anticipate no foreseeable negative societal impacts of this work.

# C  Examples of Found Theorem by POETRY

## C.1  Examples in miniF2F Dataset

```
theorem amc12b_2020_p6:
  fixes n :: nat
  assumes h0: "9 \<le> n"
  shows "\<exists>x::nat. (real_of_nat x)^2 = (fact (n + 2)
              - fact (n + 1)) / fact n"
proof -
  from assms
  show?thesis
    unfolding power2_eq_square
    by (intro exI[of _ "n + 1"]) (auto simp: field_simps)
qed
```

```
theorem mathd_algebra_422:
  fixes x :: real and \<sigma>::"real \<Rightarrow> real"
  assumes "bij \<sigma>"
    and \<sigma>:"\<forall> x. \<sigma> x = 5 * x - 12"
    and "\<sigma> (x + 1) = (Hilbert_Choice.inv \<sigma>) x"
  shows "x = 47 / 24"
proof -
  from assms
  have "bij \<sigma>"
    by (auto intro!: bijI simp: bij_def)
  show?thesis
  proof (rule ccontr)
    assume "x \<noteq> 47/24"
    thus False
      using assms
      by (subst (asm) bij_inv_eq_iff) auto
  qed
qed
```

```
theorem mathd_algebra_441:
  fixes x :: real
  assumes "x \<noteq> 0"
  shows "12 / (x * x) * (x^4 / (14 * x)) * (35 / (3 * x)) = 10"
proof -
  from assms
  show?thesis
    apply (simp add: divide_simps)
    apply algebra
    by (simp add: power4_eq_xxxx power2_eq_square)
qed
```

```
theorem mathd_algebra_487:
  fixes a b c d :: real
  assumes "b = a^2"
    and "a + b = 1"
    and "d = c^2"
    and "c + d = 1"
    and "a \<noteq> c"
  shows "sqrt ((a - c)^2 + (b - d)^2)= sqrt 10"
proof (rule real_sqrt_unique)
  show "(sqrt 10)\<^sup>2 = (a - c)\<^sup>2 + (b - d)\<^sup>2"
  proof -
    let?r = real_of_rat
    show?thesis
    proof (rule power2_eq_imp_eq)
      show "((sqrt 10)\<^sup>2)\<^sup>2 = ((a - c)\<^sup>2 + (b - d)\<^sup>2)
          \<^sup>2"
      proof -
        from assms
        show?thesis
          unfolding power2_eq_square
          apply simp
          apply (auto simp: field_simps)
          by sos
      qed
    qed (auto simp: algebra_simps)
  qed
qed (simp add: power2_eq_square)
```

## C.2  Examples in PISA Dataset

```
lemma rev_morphs: "two_binary_morphisms (rev_map g) (rev_map h)"
proof
  show "rev_map g (u \<cdot> v) = rev_map g u \<cdot> rev_map g v" for u v
  proof (simp add: rev_map_def)
    show "rev (g (rev v \<cdot> rev u)) = rev (g (rev u)) \<cdot> rev (g (rev
        v))"
      using swap
      by (simp add: g.morph)
  qed
  show "rev_map h (u \<cdot> v) = rev_map h u \<cdot> rev_map h v" for u v
  proof (simp add: rev_map_def)
    show "rev (h (rev v \<cdot> rev u)) = rev (h (rev u)) \<cdot> rev (h (rev
        v))"
      using swap
      by (simp add: h.morph)
  qed
qed
```

```
lemma lset_iterates:
  "lset (iterates f x) = {(f ^^ n) x|n. True}"
proof
  show "lset (iterates f x) \<subseteq> {(f ^^ n) x |n. True}"
  proof(cases "x \<in> lset (iterates f x)")
    case True
    thus?thesis
      by(auto simp add: in_lset_conv_lnth)
  next
    case False
    thus?thesis
      by (auto simp: in_lset_conv_lnth)
  qed
  show "{(f ^^ n) x |n. True} \<subseteq> lset (iterates f x)"
  proof safe
    fix n
    show "(f ^^ n) x \<in> lset (iterates f x)"
    proof(induct n arbitrary: x)
      case 0
      thus?case
        by(subst iterates) simp
    next
      case Suc
      thus?case
        by(subst iterates)(simp add: o_def funpow_swap1)
    qed
  qed
qed
```

```
lemma neg_distr_cond_bset_eq: "neg_distr_cond_bset (=) (=) tytok"
  unfolding neg_distr_cond_bset_def
  apply(rule predicate2I)
  apply transfer
  subgoal for A B
      apply(rule bexI[where x=B])
    subgoal
      apply safe
      subgoal
        unfolding rel_set_OO
        by(auto simp add: rel_set_def OO_def)
      subgoal
        unfolding rel_set_OO
        by(auto simp add: rel_set_def OO_def)
      done
    by(simp)
  done
```

```
lemma frag_cmul_distrib: "frag_cmul (c+d) a = frag_cmul c a + frag_cmul d a"
proof -
  show?thesis
  proof (rule poly_mapping_eqI)
    fix x
    show "lookup (frag_cmul (c + d) a) x = lookup (frag_cmul c a + frag_cmul
        d a) x"
    proof (cases "x \<in> keys a")
      case True
      thus?thesis
        unfolding lookup_add
        using lookup_frag_cmul
        by (auto simp: algebra_simps)
    qed (auto simp: in_keys_iff lookup_add in_keys_iff)
  qed
qed
```

```
lemma SETId: assumes "x |\<in>| X" shows "(Id SET X) |@| x = x"
proof -
  have "x \<in> Obj (Op SET)"
    using assms
    apply (simp add: OppositeCategory_def)
    by(simp add: SET_def SET'_def MakeCat_def)
  thus?thesis
  proof -
    assume 1: "x \<in> obj\<^bsub>Op SET\<^esub>"
    show?thesis
    proof(simp add: SET_def)
      show "id\<^bsub>MakeCat SET'\<^esub> X |@| x = x"
      proof(cases "x |\<in>| X")
        case True
        thus?thesis
          apply(simp add: SET'_def)
          apply (simp add: MakeCat_def)
          by(rule ZFfunApp)
      qed (simp add: assms)
    qed
  qed
qed
```

```
lemma (in encoding_wrt_barbs)
    indRelRSTPO_impl_SRel_and_TRel_weakly_reflect_barbs:
  fixes SRel :: "('procS \<times> 'procS) set"
    and TRel :: "('procT \<times> 'procT) set"
  assumes reflection: "rel_weakly_reflects_barbs (indRelRSTPO SRel TRel) (
      STCalWB SWB TWB)"
  shows "rel_weakly_reflects_barbs SRel SWB"
    and "rel_weakly_reflects_barbs TRel TWB"
proof -
  have "rel_weakly_reflects_barbs SRel SWB \<and> rel_weakly_reflects_barbs
      TRel TWB"
  proof (rule conjI)
    show "rel_weakly_reflects_barbs SRel SWB"
      using reflection rel_with_source_impl_SRel_weakly_reflects_barbs[where
                      Rel="indRelRSTPO SRel TRel" and SRel="SRel"]
      by (simp add: indRelRSTPO.source[where SRel="SRel" and TRel="TRel"])
    show "rel_weakly_reflects_barbs TRel TWB"
      using reflection rel_with_target_impl_TRel_weakly_reflects_barbs[where
                      Rel="indRelRSTPO SRel TRel" and TRel="TRel"]
      by (simp add: indRelRSTPO.target[where SRel="SRel" and TRel="TRel"])
  qed
  thus "rel_weakly_reflects_barbs SRel SWB" and "rel_weakly_reflects_barbs
      TRel TWB"
    by simp_all+
qed
```

