# OpenReview forum: "Proving Theorems Recursively"
_NeurIPS.cc/2024/Conference — NeurIPS 2024 poster_

### Official Review · Reviewer_p8XS · 2024-06-13

**Soundness:** 4
**Presentation:** 4
**Contribution:** 4
**Rating:** 7
**Confidence:** 4

**Summary:**

This paper designs a novel hierarchical search algorithm (POETRY) for generating formal proofs with large language models step-by-step. In particular, POETRY will first search for proof steps with proof level 0 (these steps typically correspond to subgoals in the proof), and check the correctness of the level 0 proofs by assuming that all the subgoals can be proved. If and only if the level 0 proofs are correct, POETRY will recursively search for proofs to each of the proposed subgoals. Compared with the baseline best-first search methods with the same compute, POETRY significantly improves the pass@1 succ rate on both miniF2F valid and test set, as well as the PISA test set.

**Strengths:**

-	The POETRY algorithm is neat and novel by mimicking how human write mathematical proofs hierarchically.
-	This paper is well written and easy to follow.
-	The POETRY algorithm has potentials to be further improved by incorporating premise-selection techniques such as sledgehammer or Magnushammer.

**Weaknesses:**

-	From Table 1, it seems to me that the improvement from the search algorithm is less significant than the beam search. A drawback of the beam search method is that the algorithm becomes deterministic, meaning that generating more samples per theorem does not improve its performance. Since this paper only shows pass@1 results, it is unclear how the POETRY algorithm scales with more computing resources.

**Questions:**

-	In the example shown in Figure 4, it seems to me that Path 1 is quite similar to the first part of the proof found by POETRY. Can the authors elaborate on why is the GPT-f baseline tries to utilize a more complex way to prove the first property?

**Limitations:**

The authors adequately addressed the limitations and potential negative societal impact.

---

> ### Author Rebuttal · Authors · 2024-08-06
>
> Thank you for your detailed and constructive comments. And thank you for your acknowledgment in POETRY. We hope our responses and rebuttal materials (RM) address your concerns.
>
> ## Weaknesses
> ### w1. How POETRY scales with more computing resources.
> Indeed, a large portion of the improvement comes from using the beam-search decoding method, resulting in deterministic outcomes. However, we believe the performance improvement brought by POETRY is unrelated to the specific decoding method used. Several alternative methods, such as beam sampling or sampling with a larger number of samples per input, can replace beam-search decoding.
>
> Regarding scaling POETRY with more computing resources, it can be extended with larger models, more training data, and expert iteration processes. Arming POETRY with more capable language models, the advantage of recursive theorem proving will be amplified as we tackle more complex problems that require more structural reasoning.
>
> Moreover, POETRY opens up new avenues for parallel computing. The current version of rBFS is single-threaded, adapted from BFS to showcase the performance of recursive theorem proving. In future work, we could explore parallel recursive Monte Carlo tree search, where each sub-level proof sketch can be explored concurrently. Information from different sub-searches can backpropagate to the main branch, dynamically allocating computing resources to more promising sub-searches.
>
> Therefore, we believe POETRY’s advantage will not diminish as computing resources scale up; instead, it will show more prominent advantages compared to step-by-step approaches. We are excited to explore this aspect in our future work.
>
> ## Question
> ### Q.1 Cause of complex proof in Figure 5(b).
> The GPT-f Baseline example in Figure 4(b) uses a standard best-first search algorithm to find the proof, where the choice of nodes to explore is determined by the cumulative log probability of the tactics. Although the GPT-f Baseline manages to produce the first two steps in Path 2, the log probability for the second line `show "⋀n. deg R f < n ⟹ n_mult f n = 0"` is too low compared to other generated tactics. Consequently, the search algorithm keeps exploring other states (like steps in Path 1) with higher log probabilities.
>
> While one might use a separate value function model to re-rank the proof states, it is still challenging to ensure the value function will always prefer the better path. POETRY, on the other hand, takes advantage of the Isabelle system, allowing proofs to be validated in advance and thus reducing the dependency on the value function.

---

### Official Review · Reviewer_9pow · 2024-07-12

**Soundness:** 3
**Presentation:** 3
**Contribution:** 3
**Rating:** 5
**Confidence:** 4

**Summary:**

This paper proposes POETRY, a method for formal theorem proving using language models by training the model to iteratively decompose the problem into sketches, recursively. The authors focus on Isabelle. At each step, POETRY takes a proof state and goal and predicts either a formal sketch (a proof using sorry at each step), or a ground proof step (e.g. 'by ...') that requires no recursion. These intermediate states are visited within best-first search, where the score of a node is given by the log-probability of all predictions made so far to get to that node. Intuitively, POETRY works by recursively generating lower level steps / sketches, until finding a complete proof, getting feedback from the formal environment at each step. To train the LM, the authors introduce a simple method to decompose existing Isabelle proofs from the AFP as if they had been generated by this recursive proof generation process. Experiments on minif2f show improvements on top of GPT-f and a version of Thor without Sledgehammer.

**Strengths:**

The paper is well motivated and tackles a timely topic, using a standard, hard benchmark (minif2f) for methods in this space. The writing is mostly clear (though see some notes below).

POETRY is a simple, sound and novel method to structure the proof generation process. It should be adaptable to other interactive theorem provers with some work. POETRY allows the prover model to get more intermediate feedback from the environment compared to methods that try to produce the whole proof at once. It also uses this feedback in a way that is complementary to proof repair methods (generally what the standard when we consider using intermediate feedback).

**Weaknesses:**

The choice of baselines (for Table 1) seems a bit convoluted. In particular, I don't really understand why use Thor without sledgehammer. The main point of Thor is to learn when to use hammers in proofs. Removing this makes the model much more similar to the GPT-f baseline.

As for the choice of pass@1, even though POETRY only makes a single prediction at the end, it gets feedback from Isabelle at each node in its tree. So that doesn't seem like a fair comparison either, if POETRY makes many intermediate predictions and calls Isabelle during its search, whereas GPT-f and Thor w/o sledgehammer seem to only produce and test a single prediction. It might be more fair to match the methods based on some other metric, like number of tokens generated, or number of calls to Isabelle (whichever seems to be the most significant bottleneck).

The question of "Can POETRY find longer proof?" is a bit ill posed as is. It would be possible for a method to find very long proofs that do not *need* to be long, and do better on this analysis without really being able to prove more complex theorems. What I think the authors are trying to show here is that POETRY can solve harder problems, estimating hardness by looking at proof length. For this, you might want to compare success rate based on the length of the ground truth proof: perhaps the baselines perform very poorly on theorems where the human proof is longer, whereas POETRY might have a better success rate. Another option is to show that either POETRY generates proofs of similar length to the ground truth proof (so, when POETRY generates long proofs, you'd estimate that the ground truth proof would also be long), or that it generates proofs of similar length to the baselines in cases where they all manage to prove a theorem. Any of these would help show that this result is not trivial.

**Questions:**

* Is an intermediate sketch valid (accepted by Isabelle) as long as it's syntactically correct and the last step shows the thesis? Or do you manage to get richer feedback besides that the last step declares to show the thesis?
* For problems that both POETRY and the GPT-f baseline solve, does POETRY tend to generate longer proofs?
* As for the relationship with LEGO-Prover, you mention that in some cases it is impossible to decompose a proof into lemmas, but still possible to decompose it into sketches recursively. Do you have an example?
* What exactly is the search algorithm used in the Thor w/o sledgehammer and GPT-f baselines? It is a one-shot prediction? Or do you use the (also best-first) search method described in the original GPT-f paper?
* What is Thor without sledgehammer? It sounds like a different thing other than Thor.
* I'm confused by what Figure 5 is trying to show. Fundamentally, reorganizing a proof into sketches shouldn't change its inherent complexity (e.g., the atomic proof steps). Is this just comparing the full proof length against the number of steps in the top-level sketch (without considering the deeper sketches recursively)? If you were to consider the steps in the deeper sketches recursively, I'm assuming you would not expect to see a reduction (if you do, where would that come from? do you have an example?)

**Limitations:**

Yes, adequately addressed.

---

> ### Author Rebuttal · Authors · 2024-08-06
>
> Thank you for your detailed and constructive comments. We hope our responses and rebuttal material address your concerns.
>
> ## Clarifications on the summary
> There are a few misunderstandings in summary that we would like to clarify: Within each proof sketch, POETRY operates step-by-step, searching for a complete proof sketch. At each step, POETRY takes a proof state (goal) as input and predicts one proof step, not the entire proof sketch. This step could be a conjecturing step followed by “sorry” (e.g., have c1:"a=1" sorry), or a normal proof step (e.g., by …). The search for the current sketch only stops after the sketch is complete and accepted by Isabelle.
>
> ## Weaknesses
> ### w1. Clarification on the baseline
> Both `Thor w/o Sledgehammer` and the `GPT-f Baseline` are reproductions of the GPT-f paper in the Isabelle formal system. `Thor w/o Sledgehammer` is directly adopted from the original Thor paper as an ablation setting using only LM. Since Thor is not open-sourced, we reproduced it as `GPT-f Baseline`. The methodology in `Thor w/o Sledgehammer` is the same as in our `GPT-f Baseline`, with only implementation differences detailed in Section 4.1. So the use of `Thor w/o Sledgehammer` in Table 1 is not convoluted, as it illustrates the performance of the previous paper’s implementation result of the `GPT-f Baseline`.
>
> ### w2. Pass@1 metric
> POETRY, `GPT-f Baseline`, and `Thor w/o Sledgehammer` all perform step-by-step searches using the best-first search algorithm. Therefore, the number of interactions with Isabelle and the language model are the same for POETRY and both baselines. So it’s perfectly fair to use the pass@1 score for comparison.
>
> ### w3. On "Can POETRY find longer proof?"
> It is crucial to demonstrate POETRY’s ability to find longer proofs, as it shows its capability to solve complex problems. Previous step-by-step methods often result in significantly shorter proofs than those found by POETRY, diminishing the possibility of solving more complex problems.
>
> We appreciate your insightful suggestions for analyzing POETRY’s results. We further included histograms showing the number of problems solved by GPT-f Baseline and POETRY, categorized by the length of the ground truth proofs (Figure 1 in the RM). From the Figure, it is evident that POETRY has an obvious tendency to solve harder problems (longer ground truth proofs). And outperforms the GPT-f Baseline across various problem difficulties in the multi-level subset. Therefore, we believe POETRY demonstrates a clear advantage over GPT-f Baseline and is capable of solving more complex problems.
>
> ## Question
> ### q1. The condition of a sketch being valid
> When a sketch is accepted by Isabelle (receiving ONE signal `no goals`), it means the sketch is both syntactically and semantically correct. There is no difference in the signal given back from Isabelle whether a proof sketch is valid or the complete proof is valid. POETRY relies on post-checking to determine if the proof contains `sorry`.
>
> ### q2. Proof length differences
> According to w3, POETRY can indeed solve harder proofs. For proof lengths, POETRY does not tend to generate longer proofs than the GPT-f Baseline for problems they both solve. Algorithmically, POETRY behaves almost identically to GPT-f Baseline, only proceeding to deeper levels when a complete proof sketch is found, which is challenging. Statistically, 82.3% of the problems solved by both have the same proof length, and 96.0% have proof length differences smaller than 3, caused by algorithmic randomness.
>
> Occasionally, POETRY generates longer proofs with redundant steps, as shown in Figure 2(a) in the RM. This is due to POETRY’s greedy exploration mechanism, which sometimes explores dummy sketches. These cases are rare (2.4% of solved problems where POETRY’s proof is 3 steps longer than GPT-f Baseline’s). We believe this issue can be addressed by implementing a value function to prioritize informative sketches over redundant ones in future work.
>
> Therefore, combining the w3 response, it is evident that POETRY not only handles simple problems but also excels at solving harder problems and conducting complex structural reasoning.
>
> ### q3. Example in LEGO-Prover
> Here is an example from the LEGO-Prover GitHub:
> ```Isabelle
> theorem aime_1983_p9:
>   fixes x::real
>   assumes "0<x" "x<pi"
>   shows "12 \<le> ((9 * (x^2 * (sin x)^2)) + 4) / (x * sin x)"
> proof -
>   define y where "y = x * sin x"
>   have "12 \<le> (9 * y^2 + 4) / y"
>   proof -
>     have c0: "y > 0"
>       by (simp add: assms(1) assms(2) sin_gt_zero y_def)
>     have "(9 * y^2 + 4) \<ge> 12 * y"
>       by sos
>     then show ?thesis
>       using c0 by (simp add: mult_imp_le_div_pos)
>   qed
>   then show ?thesis
>     by (simp add: power_mult_distrib y_def)
> qed
> ```
> Conjectures like `have "12 \<le> (9 * y^2 + 4) / y"` or `have c0: "y > 0"` are relatively local. Making these into independent lemmas requires many local variables, resulting in redundant proof lines and lemma statements. This example is not cherry-picked and this problem is common in many theorems. POETRY addresses this by building proofs level-by-level directly within the proof, avoiding redundant lemma statements.
>
> ### q4. The search algorithm used in the baselines
> `Thor w/o sledgehammer` and `GPT-f Baseline` use the best-first search algorithm described in the original GPT-f paper.
>
> ### q5. Explanation with Thor w/o sledgehammer
> Please refer to w1.
>
> ### q6. Confusion in Figure 5(b)
> This figure compares the full proof length (`Original`) against the number of steps in all sketches (`Recursive`), including both top-level and deeper sketches. The Original line counts each complete theorem as a single data point (e.g. proof with length 20), whereas the Recursive line treats each decomposed proof sketch as a single data point (e.g., decomposed sketches with length 9 and 11). Therefore, the total number of data points for these two plots is different. We will provide a more detailed explanation in our final paper.

---

> ### Author Response · Authors · 2024-08-10
> **Waiting for further discussion**
>
> Dear Reviewer 9pow,
>
> We hope our rebuttal sufficiently addressed your concerns. Is there any additional information we can provide that might lead you to increase your rating? We look forward to your feedback.
>
> Many thanks,
> Author

---

> > ### Comment · Reviewer_9pow · 2024-08-10
> >
> > I thank the authors for the clarifications on the method, as well as the new analyses. I do think these clarify the results, and have alleviated most of my concerns. I have raised my score assuming the new results will make it to the paper: I think the method contributes a simple but neat idea for the problem.
> >
> > About q1. (condition of a sketch being valid), I think your answer confirms my understanding, but I'd still like to clarify just because I'm interested (and this would also be helpful to clarify in the paper). Since sorry can prove any goal, my question was whether the semantic validation that Isabelle is capable of doing is essentially just that the sketch ends by concluding the thesis. For instance, this would be a toy example based on the LEGO-Prover example above:
> >
> > theorem aime_1983_p9:
> >   fixes x::real
> >   assumes "0<x" "x<pi"
> >   shows "12 \<le> ((9 * (x^2 * (sin x)^2)) + 4) / (x * sin x)"
> > proof -
> > have c0: "x = 0" by sorry
> > then show ?thesis by sorry
> > qed
> >
> > I'm understanding this would be marked by Isabelle as a valid sketch, or is there something else I'm missing?
> >
> > What I wanted to understand is exactly how much semantic feedback you can get at the sketch level, since sorry can prove anything. Your results show that structuring the proof search process in this way is helpful overall, but I'm just trying to understand whether it's more of a helpful bias to the LLM vs how much extra feedback it actually enables you to extract from the environment.

---

> > > ### Author Response · Authors · 2024-08-11
> > > **More explanation on `show ?thesis`**
> > >
> > > Dear reviewer 9pow
> > >
> > > Thank you for your prompt reply and for recognizing the value of our work. We will include all the new results in our paper.
> > >
> > > Regarding the example you mentioned, POETRY does validate the sketch, and the usefulness of the skipped conjectures was not tested. We are unable to extract additional semantic information from the environment to assist in this situation.
> > >
> > > We acknowledge this problem during the development of POETRY, and to address it, POETRY employs the “last unsolved sorry” strategy during proof search (as detailed in Appendix A.1). This strategy focuses on the final sorry in the proof sketch (i.e., the sorry in `show ?thesis sorry`), allowing for quicker validation of the proposed conjectures by first concentrating on the final incomplete part of the proof.
> > >
> > > Another potential approach could involve explicitly disallowing the use of sorry after `show ?thesis` and requiring the model to find a complete proof within the `show ?thesis` block. However, this is acceptable for proofs that can be accomplished with a single level of `show ?thesis` (i.e., `show ?thesis by ...`). For proofs requiring more complex structural reasoning within the `show ?thesis` part, this method would reduce POETRY to a step-by-step approach and fail to leverage the structural reasoning capabilities that POETRY is designed to utilize.
> > >
> > > Given the current constraints in extracting semantic information from the environment, beyond the “last sorry” strategy, a promising direction might involve using neural-based value functions as assistants to help distinguish between useful and less useful sketches. We are excited to explore this aspect in our future work.
> > >
> > > Have our responses above adequately addressed your concerns? Is there any additional information we can provide to persuade you to improve your rating?

---

### Official Review · Reviewer_BhWC · 2024-07-12

**Soundness:** 2
**Presentation:** 3
**Contribution:** 3
**Rating:** 5
**Confidence:** 4

**Summary:**

The authors introduce a method called POETRY (proving theorems recursively) for constructing formal proofs in Isabelle/HOL. POETRY performs best-first search on proof sketches guided by a language model fine-tuned on proof sketches. POETRY outperforms other algorithms guided by language models that prove theorems step-by-step. POETRY also outperforms other methods that integrate automated theorem provers and language models.

**Strengths:**

While the idea of a proof sketch is not novel, the combination of the data curation process to enable the construction of proof sketches is. This takes a step towards generating conjectures which would be crucial to making progress on neural theorem proving.

**Weaknesses:**

1. It seems to me that the real reason for the success of POETRY is not the algorithm per say, but the data curation to construct proof sketches. In this vein, it would be instructive to have a before sorry and after having sorry to illustrate how the dataset is constructed.
2. There should be more context explaining how to compare the Lean results against Isabelle/HOL. These are two different formal systems, with different proof methodologies.
3. More details on success cases and failure cases would help understanding the pros and cons of the approach taken in POETRY. For instance, are there certain kinds of problems that POETRY performs well on, e.g., geometry problems? How does POETRY perform when existentials need to be instantiated? Is it the case that POETRY can prove the same theorems as previous step-by-step approaches and can additionally prove more theorems that are longer, or do the approaches prove different short theorems?

**Questions:**

1. The distinction between a proof sketch and the decomposition of a theorem into a tree of conjectures needs to be addressed. Is there any difference?
2. In your training procedure, do you fine-tune on any theorems in the miniF2F dataset?

**Limitations:**

Yes.

---

> ### Author Rebuttal · Authors · 2024-08-06
>
> Thank you for your detailed and constructive comments. We hope our responses and rebuttal material (RM) address your concerns.
>
> ## Clarification on strengths.
> There are a few misunderstandings regarding our strengths that we would like to clarify. While we are not the first to use the term `proof sketch`, there are significant differences between the `proof sketch` in POETRY and those in previous works like `Draft, Sketch and Prove` or `LEGO-Prover`:
>
> - In previous work, `proof sketches` do not use the keyword `sorry` to defer the proof of concrete conjectures but instead use Sledgehammer to directly prove these conjectures.
> - The “proof sketch” is not used recursively in previous work, with only one sketch per problem. In the appendix, we present examples of proofs that require multiple layers of recursive expansion, which are challenging to resolve using automated tools like Sledgehammer within a single sketch. Our method allows these subgoals to be addressed incrementally, maintaining a consistent level of difficulty throughout the process.
>
> ## Weaknesses
> ### w1. Dataset construction.
> Please see Figure 1 in the paper for the process of data construction. Figure 1(a) shows the original proof, and Figure 1(b) depicts the decomposed proof sketches with sorry added. Additionally, Table 1 in the RM shows the final training crops for this problem. The final training data for POETRY and GPT-f Baseline have the same input text and the same number of training examples. The only difference is the additional `sorry` keyword in proof steps that state conjectures or subgoals. Therefore, the model trained with POETRY data does not receive more information compared to the `GPT-f Baseline`. The success of POETRY not only comes from the data curation process but also from the novel recursive BFS and the recursive proving methodology itself. We will include this table in our paper for better illustration.
>
> ### w2. Context explaining the comparison with Lean result against Isabelle.
> The results for Lean and Isabelle are indeed not directly comparable, and we will add context explaining this matter in the paper. Here is a brief explanation:
> - We follow previous work like [Li et al](https://arxiv.org/abs/2404.09939v1). to provide a comprehensive demonstration of the benchmark results in miniF2F.
> - Although the results are not directly comparable, the high-level ideas of these approaches share many similarities. The GPT-f Baseline shows much resemblance to PACT, FMSCL, and LeanDojo. By showing these results side by side, we can better understand how different formal systems perform in the benchmark.
>
> ### w3. Details on Pros and Cons of POETRY
> We have followed your advice and conducted a more thorough analysis of POETRY.
>
> - **Problem types that POETRY excels at.** In the RM, Figure 1 shows the number of problems solved by GPT-f Baseline and POETRY based on the length of the ground truth proofs. POETRY has a clear tendency to solve harder problems with longer ground truth proofs. As a domain-agnostic method, POETRY may not demonstrate a clear advantage in specific mathematical domains. However, Figure 1(c) clearly shows that POETRY has a significant advantage in problems requiring multiple levels of reasoning.
>
> - **POETRY performance on existential instantiation.** In Isabelle, existential instantiation can be performed using the `define`, `let`, or `obtain` tactics. `define` and `let` typically require explicit construction of a term that satisfies the existential condition and does not introduce a new level, making POETRY’s behavior similar to GPT-f Baseline. In contrast, obtain allows extracting a term that satisfies a given property without explicit construction, with POETRY using sorry to skip the verification of satisfiability. This approach offers more flexibility, enabling the proof to focus on variable usage and deferring validation. Although not directly involving existential instantiation, below is an example of a proof by POETRY using `obtain`:
>
>   ```isabelle
>   lemma terminates_tl_raw:
>     assumes "terminates g"
>     shows "terminates (tl_raw g)"
>   proof
>     fix st :: "bool \<times> 'a"
>     obtain n s where "st = (n, s)"
>       by (cases st) blast+
>     from assms have "s \<in> terminates_on g"
>       by (metis terminatesD)
>     thus "st \<in> terminates_on (tl_raw g)"
>       unfolding \<open>st = (n, s)\<close>
>       apply(induction s arbitrary: n)
>       by(case_tac [!] n)(auto intro: terminates_on.intros)
>   qed
>   ```
>   Here, the top-level sketch instantiates n and s and proceeds with proving conjectures, with the actual verification of the condition deferred.
>
> - **POETRY indeed proves more theorems that are longer and harder.** From Figure 1 in the RM, it is clear that POETRY is capable of proving more difficult problems (with longer ground truth lengths) and excels in multi-level subsets.
>
> Figure 2 in the RM provides more cases to better understand POETRY’s pros and cons. As illustrated, though happening sparsely, POETRY’s greedy exploration mechanism might lead to finding proofs with redundant steps or failing to find shallow proofs. However, we believe this issue can be addressed by implementing a value function to prioritize informative sketches over redundant ones in future works.
>
> ## Questions
> ### q1. Relations between proof sketches and the tree of conjectures.
> Proof sketches strictly include the tree of conjectures and also contain various other elements. For example, the obtain and show statements instantiate variables, and the subgoal tactic focuses on specific goals (shown in Figure 2(c) in the RM). POETRY inserts sorry whenever a proof step increases a proof level, allowing any form of proof sketch as long as it is permitted by the Isabelle language.
>
> ### q2. Finetuning on miniF2F.
> No, we do not finetune any data with the miniF2F dataset. All the training data we use comes from the AFP Library and Isabelle built-ins.

---

> ### Author Response · Authors · 2024-08-10
> **Waiting for further discussion**
>
> Dear Reviewer BhWC,
>
> We hope our rebuttal sufficiently addressed your concerns. Is there any additional information we can provide that might lead you to increase your rating? We look forward to your feedback.
>
> Many thanks,
> Author

---

> ### Comment · Reviewer_BhWC · 2024-08-10
> **Thank you for your response**
>
> Thank you for your response and clarifying the strengths. Perhaps I mis-worded the section on the strengths poorly, but I understood that the proof sketches both insert a sorry and are applied recursively. I'm merely pointing out that recursive decomposition and lazy evaluation is in general not novel. On the contrary, I think it is clever to take a formal proof dataset and algorithmically insert sorry/admitted at strategic points and show that this additional signal can be leveraged is novel.
>
> w1. Dataset construction.
> Thank you for the reference to Figure 1. To clarify my original question concerning having a before and after sorry, which is partially shown in Figure 1, is precisely what information is contained in the arrow from the upper level to a recursive level. In particular, does it contain the minimal proof context for the entire theorem required to make the subgoal well-typed or just the local subgoal plus a reference to a location in the existing proof? It was not clear to me which it was from the Figure or the paper description.
>
> > Therefore, the model trained with POETRY data does not receive more information compared to the GPT-f Baseline.
>
> I disagree with this statement. Your approach augments the dataset with sorry which precisely indicates when multi-level proofs are required as per your dataset curation process. Put another way, sorry indicates that a hammer will likely fail. This is additional information that other methods do not see and could benefit from! To test this, you could change the dataset curation process to insert a sorry every n proof levels instead of 1, or at random proof levels, to tease apart the effect of dataset augmentation on your approach.
>
> w2. Context explaining the comparison with Lean result against Isabelle.
> Thank you for this additional clarification.
>
> w3. Details on Pros and Cons of POETRY
> Thank you for the additional work in the RM. With regards to the types of problems that Poetry excels at, I buy that it does indeed prove longer theorems. However, I was wondering how it performs on kinds of problems such as algebraic vs. geometric, since algebraic are more compute heavy (longer proofs) and geometric problems require more constructions (i.e., existentials). Thank you for the discussion on existentials as this is a limitation of this work.
>
> I appreciate the work overall and find the approach with dataset augmentation with sorry is valuable. I am inclined to maintain my score since I still have slight concerns about weakness 1.

---

> ### Author Response · Authors · 2024-08-11
> **Response to Reviewer BhWC**
>
> Dear Reviewer BhWC,
>
> We appreciate your prompt reply and recognition of our work.
>
> ## w1. Dataset Construction
>
> Regarding your question, the language model perceives only the local subgoal when entering the next level, without knowing the location of the `sorry`. The rBFS handles all movements, whether going deeper or jumping backward. As the model progresses to the next level (as indicated by the arrow in Figure 1), the rBFS identifies the sorry and extracts the proof state immediately preceding it (i.e., the state of the conjecture). This proof state is then used to prompt the model for the next steps. Therefore, there is no difference between the input in the middle of the sketch and the input that starts a low-level sketch. While adding minimal proof context from the upper level could improve POETRY’s performance (and we believe it would), we did not include this to ensure a fair comparison with the GPT-f baseline. This allows us to clearly assess the impact of recursive proving.
>
>
> >  This is additional information that other methods do not see and could benefit from!
>
> We agree that the model receives more information compared to the GPT-f baseline in this aspect, and we acknowledge the value of the suggested ablation experiment. However, due to the limited time remaining during the rebuttal period, we may not be able to complete this experiment. We will include these ablation results in our final paper.
>
>
> ## w3. Types of problems that POETRY excels
> To provide more clarity on the types of problems where POETRY excels, we’ve included the table below, comparing the performance of POETRY and the GPT-f baseline across different mathematical categories in the PISA dataset. The results are categorized based on the directories of lemmas in the AFP library. From the table, we can see that POETRY outperforms the GPT-f baseline in most categories, including geometry and algebra.
>
> | Category | GPT-f Baseline | POETRY |
> | -------- | -------- | -------- |
> | **Graph Theory**     | 38.7%     | **41.9%**  |
> | **Cryptography**     | 65.2%     | **69.5%** |
> | **Logic**                    | 48.1%        | **53.2%**    |
> | **Linear Algebra**           | 39.3%        | 39.3%        |
> | **Set Theory**               | 54.5%        | **59.1%**    |
> | **Computation Theory**       | 51.7%        | 51.7%        |
> | **Probability and Statistics**| 52.2%       | **65.2%**    |
> | **Differential Equations**   | 52.3%        | 52.3%        |
> | **Combinatorics**            | 55.0%        | **62.5%**    |
> | **Geometry**                 | 46.7%        | **50.0%**    |
> | **Abstract Algebra**         | 41.5%        | **50.9%**    |
> | **Algebraic Geometry**       | 53.1%        | **61.2%**    |
> | **Algorithms and Data Structures** | 59.1%   | **63.6%**    |
> | **Functional Analysis**      | **46.7%**    | 40.0%        |
> | **Number Theory**            | 35.0%        | **45.0%**    |
> | **Miscellaneous**            | **50.5%**        | 50.0%    |
>
>
> Have our responses above adequately addressed your concerns? Is there any additional information we can provide to persuade you to improve your rating?

---

### Official Review · Reviewer_eTZY · 2024-07-18

**Soundness:** 4
**Presentation:** 4
**Contribution:** 3
**Rating:** 7
**Confidence:** 3

**Summary:**

This paper introduces POETRY, a new method to prove theorems recursively. The key ideas are to use a modified best first search algorithm for the search part, and a *sorry* tactic for assumptions at the current level (to be proven later). The authors provide the intuition that this recursive structure allows POETRY to prove theorems in a top-down fashion similar to humans, getting into the details of proving a conjecture only if it is actually relevant to the best overall proof being explored. The authors conduct experiments with two standard benchmarks, showing notable improvements over baselines and SOTA search-based methods (but not LEGO-Prover etc. which rely on substantially larger general purpose LLMs).

**Strengths:**

The paper is very well structured and clearly written. Intuitions, method details, connection to existing methods, limitations, and take away messages from experiments are all very well articulated.

The idea seems simple but is apparently novel (see 'weaknesses' below, related to this).

The gains over several baselines are notable, of 5% or more (absolute).

I am assuming the authors will publicly release the code of their POETRY system for further research on this topic.

**Weaknesses:**

Not being very familiar with the area, I am surprised none of the existing SOTA methods use a similar recursive, top-down search of in theorem proving. I will have to defer to other, more knowledgeable reviewers for assessing novelty of the present work.

I did not fully follow why a *novel* recursive best-first search strategy is needed here. The description of this section (3.2) can probably use some clarification. E.g., why could one not account for the conjecture's level in the utility of the conjecture, and thus implicity enforce level-by-level proof search? On the same note, could the authors comment on the relationship between their proposed recursive best-first search and a combination of standard breadth-first search (i.e., staying within a level) and best-first search (i.e., preferring to explore the most promising node first)?

Just for completeness, it would have been good to know how well very large LLM based methods, such as LEGO-Prover, do on the considered benchmarks.

**Questions:**

Please see weaknesses section above.

**Limitations:**

Yes

---

> ### Author Rebuttal · Authors · 2024-08-06
>
> Thank you for your detailed and constructive comments. We hope our responses and rebuttal material address your concerns. As you mentioned in the strengths section, we will release all the code, models, and data on the POETRY system to support further research on this topic.
>
> ## Weaknesses
> ### w1. Novelty on the recursive method.
> To the best of our knowledge, we are the first to introduce this type of recursive, top-down search in the domain of neural theorem proving. The challenge of the step-by-step proof search has been long recognized, but no effective method has been proposed until now. Other reviewers (BhWC, 9pow, p8XS) acknowledge the novelty, simplicity, and effectiveness of POETRY’s design, and believe it will be crucial to advancing neural theorem proving.
>
> ### w2. The necessity of the novel recursive best-first search
> There are various options to handle recursive proving approaches, and we chose to use recursive best-first search (rBFS) for the following reasons:
>
> - **Alignment.** The rBFS algorithm aligns well with the recursive proving paradigm. As proofs are broken into several proof sketches, it's natural to align each sketch with an individual BFS search.
> - **Equal Treatment for Equivalent Matters.** Within a single problem, each sketch we seek to find is fundamentally the same, allowing the same structure to occur in every sketch. Thus, we want equal treatment for every sketch and avoid unnecessary defunctions at certain levels.
> - **Better extensiveness.** Equal treatment also enhances the algorithm’s extensiveness. Currently, we experiment with BFS as the core algorithm, but it will be easy to integrate new search algorithms like MCTS or other improved search methods into rBFS.
> - **Easy parallelization.** Currently, rBFS is executed in a single-threaded manner, but since each sub-search is identical to the others, rBFS can easily run in parallel, leveraging more powerful computation resources.
>
> As you pointed out, one could indeed use vanilla BFS for recursive theorem proving with a dedicated design of node priority scores to enforce implicit level-by-level search. However, while it’s straightforward to prioritize the conjecture’s level at the top level, ensuring the same behavior for the conjecture level inside the proof of the top-level conjecture becomes complex. Adding the uncertainty of log-prob makes the algorithm intricate and uncontrollable. Nonetheless, we acknowledge that such an algorithm is possible and may have advantages over rBFS in certain aspects. However, this design would lose the beneficial properties of rBFS listed above.
>
> Regarding the combination of standard breadth-first search and best-first search, it’s important to note that the current best-first search operates similarly to the breadth-first search in the GPT-f paper. The priority score for the best-first search is cumulative log-prob, where the score for each node is calculated by accumulating all the log-prob from the root node to the current node. Consequently, nodes at deeper levels tend to have lower scores, and the actual behavior of best-first search resembles breadth-first search. Therefore, we could not come up with a way to combine breadth-first search and best-first search to tackle recursive proving approaches.
>
>
> ### w3. The performance on miniF2F with large language model.
> We have included Table 2 in the RM, listing baseline methods that utilize large language models on the miniF2F benchmark. It is important to note that these methods use a completely different approach to prove theorems. Instead of performing searches using BFS algorithms, these approaches leverage the general-purpose capabilities of LLMs to translate natural language proofs into formal code (often termed autoformalization). Therefore, comparing these methods with POETRY is not only unfair in terms of model size and the amount of training data but also incomparable in core methodology.

---

### Author Rebuttal · Authors · 2024-08-06

Dear Reviewers and ACs,

Thank you very much for the time and effort you have dedicated to reviewing our paper. We appreciate the thorough suggestions and constructive feedback on our manuscript.

We are also grateful for the positive recognition from the reviewers regarding our motivation (eTZY, 9pow, p8XS), contribution (eTZY, BhWC, 9pow, p8XS), and strong results (eTZY, p8XS), as well as the potential future impact of our work (eTZY, BhWC, 9pow, p8XS). We acknowledge the concerns raised by reviewers BhWC and 9pow, which may stem from some previously incomplete observations in our work. Reviewer 9pow raised concerns about our baseline comparison, which appear to stem from misunderstandings of our methods and baseline. POETRY operates step-by-step, like GPT-f Baseline, within each proof sketch, without incurring extra computation costs compared to the baseline method, ensuring a fair comparison. We will provide further explanations in our revised manuscript.

We have carefully considered all the suggestions provided by the reviewers and included the additional information requested in our one-page rebuttal. We have denoted the attached PDF document as rebuttal material, abbreviated as RM. The key updates are as follows:

- **(BhWC)** Table 1 compares the constructed training corpus for GPT-f Baseline and POETRY, providing a clear view of the differences in training data.
- **(eTZY)** Table 2 illustrates benchmark results using a large language model. These results are not comparable but are presented for demonstration purposes.
- **(BhWC and 9pow)** Figure 1 shows a histogram of the number of problems solved by `GPT-f Baseline` and POETRY based on the length of the ground truth proofs. This analysis reveals that POETRY tends to solve more complex problems and excels at solving problems requiring structural reasoning.
- **(BhWC and 9pow)** Figure 2 provides two failure cases showcasing issues caused by the greedy exploration process, which can lead POETRY to find proofs with redundant steps or fail to find shallow proofs. However, these occurrences are sparse.

Additionally, we outline our paper's main contributions, including additional conclusions drawn during the rebuttal:

- We propose POETRY, a novel approach for neural theorem proving. Addressing a long-standing issue in previous step-by-step approaches, we introduce a recursive proving approach to find proofs hierarchically within the proof. **To our knowledge, we are the first to explore the recursive proving paradigm in neural theorem proving.**
- We introduce a novel recursive best-first algorithm that naturally aligns with the idea of recursive theorem proving. Additionally, it has good extensiveness and the potential for parallel computation.
- Experiments conducted on the miniF2F and PISA datasets demonstrate significant performance gains with our POETRY approach over state-of-the-art methods. POETRY achieves an average proving success rate improvement of 5.1% on miniF2F.
- A substantial increase in the maximum proof length found by POETRY is observed, from 10 to 26. Through thorough analysis, we confirm that POETRY is capable of solving more complex problems and excels at solving problems requiring structural reasoning.

We thank all the reviewers again for their time and effort in reviewing our paper and are committed to addressing every issue raised.

---

### Decision · Program_Chairs · 2024-09-25

**Decision:**

Accept (poster)

**Comment:**

This paper introduces POETRY, a new method to prove theorems recursively in a
a top-down fashion similar to humans, proving a conjecture only if it is actually relevant
to the best overall proof being explored. The best-first search on proof sketches is guided
by a language model fine-tuned on proof sketches. The ideas, although not revolutionary, appear to
be novel in this domain, the paper is very well written and the experiments show a significant
improvement on the state of the art.

One reviewer inscreased their score from borderline reject to borderline accept after the author-reviewer discussion.